# Inclusion of 2D Transition Metal Dichalcogenides in Perovskite Inks and Their Influence on Solar Cell Performance

**DOI:** 10.3390/nano11071706

**Published:** 2021-06-29

**Authors:** Nicola Taurisano, Gianluca Bravetti, Sonia Carallo, Meiying Liang, Oskar Ronan, Dahnan Spurling, João Coelho, Valeria Nicolosi, Silvia Colella, Giuseppe Gigli, Andrea Listorti, Aurora Rizzo

**Affiliations:** 1Dipartimento di Matematica e Fisica “E. De Giorgi”, Campus Ecotekne, Università del Salento, Via Arnesano, 73100 Lecce, Italy; nicotaurisano86@gmail.com (N.T.); gianlucabravetti93@gmail.com (G.B.); giuseppe.gigli@unisalento.it (G.G.); 2CNR NANOTEC, c/o Campus Ecotekne, Institute of Nanotechnology, Via Monteroni, 73100 Lecce, Italy; sonia.carallo@nanotec.cnr.it (S.C.); aurora.rizzo@nanotec.cnr.it (A.R.); 3School of Chemistry, Trinity College Dublin, Dublin 2, Ireland; liangm@tcd.ie (M.L.); ORONAN@tcd.ie (O.R.); SPURLIND@tcd.ie (D.S.); jcm.coelho@fct.unl.pt (J.C.); nicolov@tcd.ie (V.N.); 4CRANN and Amber, Trinity College Dublin, Dublin 2, Ireland; 5CENIMAT|i3N, Departamento de Ciência de Materiais, Faculdade de Ciências e Tecnologia Universidade NOVA de Lisboa and CEMOP/UNINOVA, Campus da Caparica, 2829-516 Caparica, Portugal; 6CNR NANOTEC, c/o Department of Chemistry, Institute of Nanotechnology, University of Bari ‘Aldo Moro’, Via Orabona 4, 70126 Bari, Italy; silvia.colella@nanotec.cnr.it; 7Department of Chemistry, University of Bari “Aldo Moro”, Via Orabona 4, 70126 Bari, Italy

**Keywords:** perovskite solar cells, MoS_2_ additive, morphology, heterogeneous nucleation

## Abstract

Organic–inorganic hybrid perovskite materials have raised great interest in recent years due to their excellent optoelectronic properties, which promise stunning improvements in photovoltaic technologies. Moreover, two-dimensional layered materials such as graphene, its derivatives, and transition metal dichalcogenides have been extensively investigated for a wide range of electronic and optoelectronic applications and have recently shown a synergistic effect in combination with hybrid perovskite materials. Here, we report on the inclusion of liquid-phase exfoliated molybdenum disulfide nanosheets into different perovskite precursor solutions, exploring their influence on final device performance. We compared the effect of such additives upon the growth of diverse perovskites, namely CH_3_NH_3_PbI_3_ (MAPbI_3_) and triple-cation with mixed halides Cs_x_ (MA_0.17_FA_0.83_)_(1−x)_Pb (I_0.83_Br_0.17_)_3_ perovskite. We show how for the referential MAPbI_3_ materials the addition of the MoS_2_ additive leads to the formation of larger, highly crystalline grains, which result in a remarkable 15% relative improvement in power conversion efficiency. On the other hand, for the mixed cation–halide perovskite no improvements were observed, confirming that the nucleation process for the two materials is differently influenced by the presence of MoS_2_.

## 1. Introduction

Extensive, ongoing research has been performed to date towards the development of highly efficient and stable perovskite solar cells (PSC) [1]. Organic–inorganic metal halide perovskites, with the general formula ABX_3_ (where A is a cation, B is a divalent metal cation and X is a halide), are a class of semiconductors that have the potential to deliver high-efficiency photovoltaic devices at mild temperature processing conditions and low cost [2]. These materials feature unique optical and electronic properties, such as high carrier mobility, long charge diffusion length, low trap-state densities and intense broadband absorption [3].

Perovskite crystalline structure, morphology, dimension, and distribution of the poly crystallite perovskite thin films are critical factors that affect the optoelectronic and photovoltaic properties [4]. Currently, one of the main challenges for PSC development is the deposition of high-quality films with optimized morphology, large grains and which are pinhole-free. Many different approaches have been investigated to improve the film quality: annealing conditions [5], compositional [6] and solvent engineering [7], interfacial modification [8], deposition methods [9], and, finally, additive engineering [10,11,12,13]. The inclusion of additives into the perovskite precursor solution is widely adopted, as their presence impacts the final morphology of perovskite films [14], stabilizes the active crystalline phase [15], tunes the energy level alignment between material constituents, and suppresses non-radiative recombination in perovskite materials [16]. Furthermore, additives play an important role in perovskite crystal growth and can lead to a stronger control over nucleation and crystallization kinetics [11], or activate surface passivation mechanisms, achieving highly crystalline perovskite films that are suitable for device integration. Several types of additives have been added into perovskite precursor solutions, including polymers [13], molecules [17], organic [18,19] and inorganic [20] halide salts, inorganic acids [21], carbon-based materials and nanoparticles [22]. Within this collection, interesting perspectives involve the use of systems characterized by extended surfaces, such as two-dimensional (2D) materials, which could extend the interaction over a region of large perovskite grains [23,24,25].

Two-dimensional transition metal dichalcogenides (2D-TMD), obtained by liquid-phase exfoliation, are characterized by high electrical conductivity and charge carrier mobility, tunable optoelectronic properties, and offer the possibility of engineering their surfaces by physical or chemical methods. Therefore, with PSC advancement in mind, 2D-TMDs offer the possibility of integration into multiple features: as electrodes, as charge transporting layers, and/or as an additive to be used in the precursor solutions of perovskite active layers [26].

Two-dimensional flakes of MoS_2_, MoSe_2_ and TiS_2_ have recently been used as an interlayer between perovskite and either the hole transporting layer (HTL) or electron transporting layer (ETL) [27,28,29]. This prevents the formation of shunt contacts between the perovskite and the electrode and provides a more suitable energy band alignment between the perovskite active layer and the transporting layer. Finally, chemically exfoliated MoS_2_ was employed both as an additional component of the perovskite precursor solution and an interfacial layer to enhance PSC [30] efficiency in the p-i-n architecture with PEDOT:PSS as the HTL [30]. The inclusion of MoS_2_ enlarged the grain size and improved the crystalline quality of MAPbI_3_ perovskite films. Furthermore, due to the incorporation of an MoS_2_ interlayer, the direct contact between hydrophilic PEDOT:PSS and MAPbI_3_ film is avoided, preventing early deterioration of device performance.

In this work, we focused on the inclusion of MoS_2_ TMD sheets into a perovskite precursor solution of two representative benchmark materials, MAPbI_3_ and mixed cation-halide CsMAFAPbIBr (MA = methylammonium, FA = formamidinium), with the aim of improving our knowledge of the role of such an additive in diverse perovskite formulations. The solar cell architecture was an inverted p-i-n structure employing organic transporting layers as sketched in Figure 1a,b. Our findings show that the integration of MoS_2_ into the MAPbI_3_ perovskite active layer leads to larger grain size, resulting in improved device performance. In fact, the PSCs show a 15% improvement in power conversion efficiency (PCE) through embedding TMDs in the perovskite active layer. Conversely, identical additive engineering for CsMAFAPbIBr perovskite did not show similar device improvements, suggesting that the polycrystalline perovskite film formation is influenced differently by the presence of 2D MoS_2_ depending on the different ions involved in solution.

## 2. Experimental Section

### 2.1. Materials

Lead (II) iodide ultradry (PbI_2_, metals basis, 99.999%) was purchased from Alfa Aesar (Kandel, Germany), methylammonium iodide (MAI) from Dyesol (Kapaklı, Tekirdağ, Turkey). Formamidinium iodide (FAI, ≥98%, powder), lead (II) bromide (PbBr_2_, 99.999%, powder), cesium iodide (CsI, 99.999%), dimethyl sulfoxide (DMSO, anhydrous, 99.9%), N,N-dimethylformamide (DMF, anhydrous, 99.8%), toluene (anhydrous, 99.8%), chlorobenzene (anhydrous, 99.8%), 2-propanol (anhydrous, 99.5%), bathocuproine (BCP, 96%), polytriarylamine (PTAA) were supplied from Sigma Aldrich (St. Louis, MO, USA). Methylammonium bromide (MABr, >99.5%, recrystallized 4 times) was purchased from Luminescence Technology Corp. (Taipei, Taiwan). Phenyl-C61-butyric acid methyl ester (PCBM) was purchased from Nano-c (Westwood, MA, USA). ITO coated glass substrates were purchased from Kintec (Hong Kong, China). All chemicals were used as received without further purification.

### 2.2. Preparation of 2D MoS_2_ Nanosheets 

2 g of MoS_2_ powder were added to 40 mL of N-methyl-2-pyrrolidone (NMP), and a probe sonic tip was used to sonicate the solution for a certain number of hours (power: 60% amplitude). The sonic tip was pulsed for 6 s on and 2 s off to avoid damage to the processor and reduce solvent heating, and thus, degradation. The beaker was connected to a cooling system that allowed for cold water (under 5 °C) to flow around the dispersion during sonication. Then, exfoliated MoS_2_ nanosheets in NMP solvent were centrifuged at high spin rate (10,000 rpm) to obtain the MoS_2_ nanoflakes. Then, these MoS_2_ nanosheets were redispersed in DMF through bath sonication in a concentration of 0.08 mg mL^−1^. This solution was then used for the different additions to perovskite precursor solutions.

### 2.3. Preparation of the Perovskite Precursors Solutions 

All the perovskite precursor solutions were prepared in a N_2_-filled glovebox. The MAPbI_3_ perovskite solutions were prepared by mixing methylammonium iodide (MAI, 159 mg) and lead (II) iodide (PbI_2_, 461 mg) in a mixture of N,N-dimethylformamide (DMF, 629 µL) and dimethyl sulfoxide (DMSO, 71 µL). The CsMAFAPbIBr perovskite precursors solutions were prepared by mixing formamidinium iodide (FAI, 172 mg), lead (II) iodide (PbI_2_, 507 mg), lead (II) bromide (PbBr_2_, 73 mg), methylammonium bromide (MABr, 22 mg) and cesium iodide (CsI, 17 mg) in a mixture of N,N-dimethylformamide (DMF, 900 µL) and dimethyl sulfoxide (DMSO, 100 µL). 2D MoS_2_ nanosheets dispersion in DMF was added into perovskite precursor solutions at different volume ratios (5%, 10%, 20%), keeping the DMF/DMSO volume ratio constant.

### 2.4. Photovoltaic Device Fabrication 

Perovskite solar cell architectures are shown in Figure 1a,b. ITO-coated glass substrates were cleaned by ultrasonication in a deionized water, acetone, 2-propanol. Polytriarylamine (PTAA) layer (1.5 mg mL^−1^ in toluene) was deposited by spin coating at 6000 rpm for 30 s and annealed at 100 °C for 10 min. The rest of the processes were performed in an N_2_-filled glovebox. The MAPbI_3_ perovskite precursor solution was spin coated onto PTAA coated ITO substrates at 4000 rpm for 25 s; for solvent dripping, 200 µL of toluene was dropped onto the film 15 s prior to the end. The CsMAFAPbIBr perovskite precursor solution was spin coated onto PTAA coated ITO substrates by a consecutive two-step spin coating process at 1000 rpm and 6000 rpm for 10 s and 20 s, respectively; for solvent dripping, 200 µL of chlorobenzene was dropped onto the film 5 s prior to the end. The perovskite film was then annealed for 10 min at 100 °C and then allowed to cool down to room temperature. Then, a PCBM solution in chlorobenzene (25 mg mL^−1^) was spin coated onto the perovskite layer at 1000 rpm for 60 s. Finally, a BCP solution in 2-propanol (0.5 mg mL^−1^) was spin coated at 6000 rpm for 20 s. Solar cell devices are completed by thermal evaporation of 100 nm Al electrodes.

### 2.5. Materials Characterization 

The scanning electron microscopy (SEM) imaging was performed by a MERLIN Zeiss SEM FEG instrument at an accelerating voltage of 5 kV, using an in-lens detector. The particle size distribution was estimated using the open-source ImageJ software, by measuring the major axis of 100 perovskite grains for each sample. HRTEM images were acquired on an FEI Titan (Thermo Fisher Scientific Inc., Waltham, MA, USA) operating at an acceleration voltage of 300 keV. The EDX spectrum of the MoS_2_ dispersion was acquired on Jeol2100 (Jeol Ltd., Akishima, Tokyo, Japan) operating at 200 keV and equipped with an Oxford Instruments 80mm^2^ silicon drift detector with a 10° holder tilt. Ultraviolet–visible (UV-Vis) absorption spectra were measured on PerkinElmer (Lambda 1050, Waltham, MA, USA) spectrophotometer in the 300–800 nm wavelength range at room temperature. Steady-state and time-resolved photoluminescence (PL) was performed using an Edinburgh FLS920 spectrometer (Edinburgh Instruments, Scotland, UK) equipped with a Peltier-cooled Hamamatsu R928 photomultiplier tube (185–850 nm). An Edinburgh Xe900 450 W Xenon arc lamp (Edinburgh Instruments, Scotland, UK) was employed as the exciting light source. Emission lifetimes were determined using the single photon counting technique by means of the same Edinburgh FLS980 spectrometer (Edinburgh Instruments, Scotland, UK) using a laser diode as the excitation source and a Hamamatsu MCP R3809U-50 as detector.

### 2.6. Photovoltaic Device Characterization 

The devices were characterized in an N_2_ atmosphere by using a Keithley 2400 Source Measure Unit (Tektronix, Berkshire, UK) and AirMass 1.5 Global (AM1.5G) solar simulator (Newport 91160A, Irvine, CA, USA) under an irradiation intensity of 100 mW cm^−2^. Current–voltage characteristics were acquired at voltages ranging from 1.2V to −0.2 V. The step voltage is fixed at 10 mV and the delay time to 100 ms.

## 3. Results and Discussion

### 3.1. Properties and Characterizations of Materials

The MoS_2_ nanosheets used in this work as an additive in hybrid halide perovskite solutions were prepared by liquid-phase exfoliation (see experimental section). Firstly, the 2D material was characterized in order to confirm an effective exfoliation. The scanning electron microscopy (SEM, Figure 2a) image shows few-layered MoS_2_ nanoflakes. High-resolution TEM (HRTEM, Figure 2b) and its corresponding fast Fourier transform (FFT, inset of Figure 2b) pattern reveal the highly crystalline structure of the as exfoliated MoS_2_ material. Moreover, the exfoliated flakes reveal well-defined edges and no apparent damage in the basal planes, suggesting that the obtained material is of high quality.

In the EDX spectrum (Figure 2c), the molybdenum (Mo) and sulphur (S) peaks are clearly present confirming the identity of the nanosheets. The carbon (C) peak is also clearly identifiable and attributable to both the lacey carbon grid, as well as to possible residual DMF solvent. Additional peaks can be attributed to the background scattered signal (iron (Fe) and chromium (Cr)) from the TEM polepieces. Copper (Cu) peaks are present due to the copper TEM grid. The silicon (Si) peak is due to an EDX spectral artefact, being most probably attributable to an internal fluorescence peak from the silicon window on the drift detector.

The as-obtained MoS_2_ sheets were dispersed in N,N-dimethylformamide and employed as an additive for MAPbI_3_ perovskite precursor solution at different volume ratios: 5, 10, 20 (%, V/V). A preliminary check on the compatibility between MoS_2_ TMD sheets and the reaction environment in which the nucleation and growth processes of the perovskite material take place confirmed the possibility of using this material as an additive for perovskite precursor solutions. In this environment, reasonably good solubility of the TMD is observed, and no precipitation or phase separation occur during the spin coating process. This high solubility can be explained by the coordinated equilibrium of lead cations and iodide ions present in the perovskite precursor solution along with MoS_2_ [31]. Indeed, the S nonbonding lone-pair orbital can function as an electron donor (Lewis base), interacting with lead cations (Lewis acids) in solution, while halide ions can strongly coordinate molybdenum [32]. The resulting films are spin coated on glass/ITO/polytriarylamine (PTAA), which is the hole transporting material employed in the solar cell architecture. These films are smooth, particularly homogeneous and pinhole-free (insets of Figure 3). The thickness of the perovskite films was found to slightly increase upon the addition of MoS_2_ sheets (Table 1). Small differences among the three samples are within experimental error.

SEM images (Figure 3b,c) show the MoS_2_ additive’s influence on the MAPbI_3_ perovskite grain size: the addition of the additive into the perovskite precursor solutions, at low–intermediate concentrations (5% and 10%), results in the formation of larger grains. The particle size analysis of these images (Figure 4, Table 2) reveals a significant increase indeed in the average grain size, from 116 nm for the pristine sample to 177 nm and 187 nm for the MoS_2_ (5%) and MoS_2_ (10%) perovskite samples, respectively. Moreover, the 10% MoS_2_ additive sample has the narrowest grain size distribution, with more than 50% of particles located in a small range from 160 nm to 220 nm. The implication of this observation is a reduced extension of the grain boundary region. This prefigures a reduced charge loss, typically occurring at the grain boundary region, giving rise to an improvement in final device performance.

There are several factors that can affect the perovskite’s morphology, including the intrinsic properties of the material itself, the deposition method, the presence of impurities, the surface energy of the substrate, and the application of post-treatments [31]. The MoS_2_ additive seems to be able to regulate film morphology by acting upon the crystal growth and by altering the colloid distribution in the perovskite precursors [31]. This results in high-quality, pinhole-free perovskite films with larger grain size and filled grain boundaries [33,34]. Since the crystal growth rate is relatively fast and is a function of solution supersaturation, reaching a high nucleation rate before the onset of crystal growth is required to improve perovskite film coverage. The heterogeneous nucleation mechanism in addition provides fewer nucleation sites, when compared to homogeneous nucleation, leading to the formation of larger crystalline domains [34,35]. This interesting observation is extended here to the deposition of perovskite onto organic PTAA substrates, showing that TMDs can also affect this kind of device, which would be ideally suited for flexible, light, and portable PSCs. Noticeably, at 20% MoS_2_ additive concentration, the effect on the grain size is lost as these films exhibit similar grain sizes and particle size distribution to pristine perovskite (Figure 4, Table 2). The effect of such high MoS_2_ concentration is ascribed to an increased number of heterogeneous nucleation sites that would impair the formation of large grains during the perovskite crystal growth. This is suggested by the SEM image in Figure 3d. A morphology characterized by smaller grains, such as the one recorded for the 20% MoS_2_ additive perovskite film, is in general associated with poor charge transport and collection in PSCs, which are limited by inter-grain boundary recombination losses [36].

The approach reported in this work was extended to mixed cation–halide CsMAFAPbIBr perovskite, which is among the best-performing perovskite material for solar cells. Surprisingly, we found that the inclusion of MoS_2_ additive into the triple-cation did not significantly affect the final morphology of the perovskite films (Figure 5), as all the samples exhibited a uniform and compact surface with similar grain size distribution (Figure 6, Table 3). This can mainly be ascribed to the marginal role of the additive in influencing and controlling the crystallization dynamics of the CsMAFAPbIBr perovskite material. We suppose that in this case, the lower-solubility cesium salts act as heterogeneous nucleation seeds, promoting heterogenic crystal growth [35].

UV-Vis absorption spectra of the perovskite thin films are shown in Figure 7a. Both pristine MAPbI_3_ and MAPbI_3_:MoS_2_ films show the same absorption onset. MoS_2_ TMD additive, as expected, does not contribute to the absorption spectra due to the high extinction coefficient of perovskite material and due to low additive concentration. Conversely, steady-state photoluminescence (PL, Figure 7b) revealed that the MoS_2_ inclusion causes PL quenching, and this is more pronounced at high additive concentrations. We also observed a very small blue shift of the PL band upon additive addition, probably due to a surface and/or grain boundary trap-states passivation effect. The PL quenching observed is a consequence of an enhanced charge transfer from the perovskite structure to the MoS_2_ TMD sheets. This is in line with what has previously been observed on alternative systems [37,38] and is related to the optimal band gap alignment and carrier dynamics for these two materials. Moreover, the mechanism of high carrier transfer efficiency among these heterostructures involves both holes and electrons, leading to an improvement in device performance [38] through an intelligently designed charge funneling mechanism. The time-resolved photoluminescence (TRPL) decay (Figure 7c) is consistent with this picture, as the charge carrier lifetime (τ_av_) [39] decreases with increasing MoS_2_ content, from 120 ns for the pristine MAPbI_3_ perovskite to 9 ns for the 20% MoS_2_ additive perovskite.

### 3.2. Photovoltaic Performances

Perovskite films with and without MoS_2_ additive were included in p-i-n solar cell architectures, as shown in Figure 1. Perovskite solar cells bearing this layout have advantages over n-i-p ones because of the possibility of low-temperature preparation and negligible J-V hysteresis effects [40]. Additionally, the reasons for the selection of PTAA as the hole transporting layer include increased resilience to the negative effects of oxygen and moisture due to its hydrophobicity, and remarkable intrinsic hole mobility [41]. Figure 8 and Table 4 display the J-V curves and the photovoltaic performances of the MAPbI_3_ material. The J-V characteristics indicate that MoS_2_ sheets as an additive improve device properties. The best PCE of 17.4% was recorded for the 10% MoS_2_ additive PSC (FF = 68.9%, V_OC_ = 1.02 V, J_SC_ = 24.76 mA/cm^2^); however, even at low concentration (5%), the inclusion of MoS_2_ additive leads to better performance (16.0%) with respect to the pristine MAPbI_3_ device (15.1%). These findings are consistent with the morphological characterization, for which low and intermediate additive concentrations enable us to obtain larger grain sizes, and therefore reduced grain boundary recombination. The J_SC_ shows no significant change with additive modification, and thus the increment in PCE is mainly related to the enhanced V_OC_. The V_OC_ improvement could be a consequence of suppressed non-radiative recombination processes within the structure, which occur in the absorber layer and at the interfaces between the perovskite and the transporting layers [42]. On the other hand, for higher MoS_2_ additive concentration (20%), the PCE significantly drops to 13.3%. Adding an excess of MoS_2_ TMD sheets can lead to the formation of a poor-quality perovskite layer, which can be attributed to aggregated MoS_2_ nanosheets impairing perovskite growth [43], resulting in shorter diffusion lengths of photogenerated carriers, higher density of trap states, and high carrier recombination rates. This is an indirect proof of how important it is for MoS_2_ sheets to be well exfoliated, and thus well isolated, rather than in a bulky or aggregated phase. Decreased performance of the 20% MoS_2_ PSC is in good agreement with the photoluminescence measurements that showed the almost completely quenched PL signal at higher additive concentrations (20%).

On the other hand, the photovoltaic performance of mixed cation–halide CsMAFAPbIBr PSCs (Figure 9, Table 5) implemented in an identical device architecture, reported in Figure 1b, did not show any significant change following the MoS_2_ inclusion. PCE of the best device was above 18%, V_OC_ was between 1.01 and 1.03 V, FF was above 78%, and J_SC_ was above 23 mA/cm^2^ for the pristine perovskite and CsMAFAPbIBr:MoS_2_ (5%). For MoS_2_ (10%) and MoS_2_ (20%), J_SC_ slightly decreases to 22.62 and 22.08 mA/cm^2^, respectively, possibly due to the occurrence of some nanosheet aggregation as observed for MAPbI_3_. It should be noted, however, that these differences are negligible if we consider a similar mean value and statistical distribution, which reflects the negligible differences in film morphologies (Figure 5) observed with and without MoS_2_.

## 4. Conclusions

In summary, we demonstrated an effective additive-engineering approach to improve the morphology of perovskite films grown on an organic substrate, leading to superior device performance. Liquid-phase exfoliated molybdenum disulfide sheets were employed as an additive in MAPbI_3_ perovskite precursor solution, to measure their effect at diverse concentrations and upon diverse precursor formulations. Our findings suggest that MoS_2_ incorporation (10% V/V) into a MAPbI_3_ perovskite photoactive layer results in high-quality films with larger grains and optimized morphology, suitable for device integration. As a result, the champion device yields a remarkable PCE of 17.4%, 15% higher than the undoped device. The beneficial impact of 2D layered materials originates from seed induced heterogeneous nucleation, resulting in superior perovskite film morphology, and therefore in high-performance PSCs. Conversely, for the mixed cation–halide perovskite, no improvements were observed with the addition of the additive to the precursor ink. Our results confirm that the nucleation process differs for distinct perovskite precursor compositions, and so, that the process is influenced differently by the presence of additives. These results contribute to the development of perovskite-based solar cells and, more specifically, to the wide research front focused on controlling photoactive film deposition and growth.

## Figures and Tables

**Figure 1 nanomaterials-11-01706-f001:**
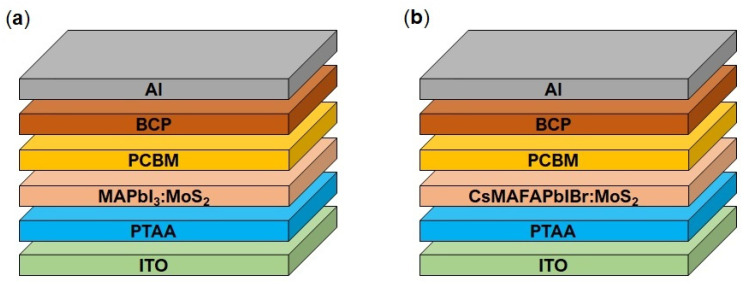
(**a**) Schematic *p*-i-n architectures of the MAPbI_3_:MoS_2_ and (**b**) CsMAFAPbIBr:MoS_2_ PSCs.

**Figure 2 nanomaterials-11-01706-f002:**
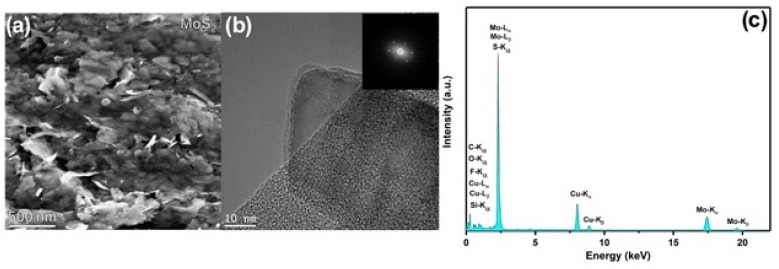
(**a**) SEM image. (**b**) High-resolution TEM images of exfoliated MoS_2_, inset in (**b**) is the fast Fourier transform pattern of exfoliated MoS_2_. (**c**) The EDX spectrum confirms the identity of the nanosheets.

**Figure 3 nanomaterials-11-01706-f003:**
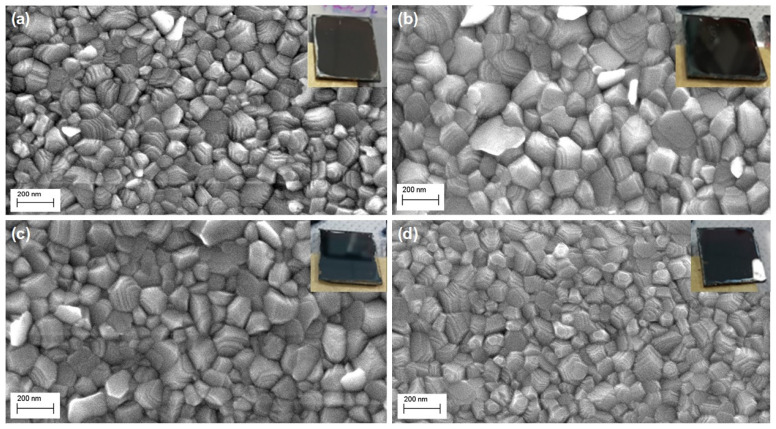
Top-view SEM images of the as-prepared MAPbI_3_ perovskite films, with and without MoS_2_ additive: (**a**) MAPbI_3_; (**b**) MAPbI_3_ + MoS_2_ (5%); (**c**) MAPbI_3_ + MoS_2_ (10%); (**d**) MAPbI_3_ + MoS_2_ (20%). In the insets are reported the photographs of the corresponding perovskite films coated on glass/ITO/PTAA substrates.

**Figure 4 nanomaterials-11-01706-f004:**
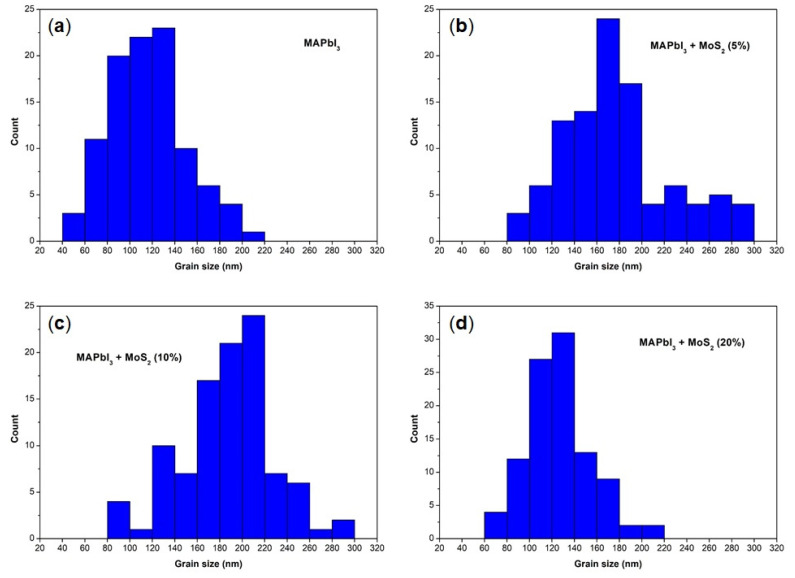
Particle size distribution histograms of the as-prepared MAPbI_3_ perovskite films, with and without MoS_2_ additive: (**a**) MAPbI_3_; (**b**) MAPbI_3_ + MoS_2_ (5%); (**c**) MAPbI_3_ + MoS_2_ (10%); (**d**) MAPbI_3_ + MoS_2_ (20%).

**Figure 5 nanomaterials-11-01706-f005:**
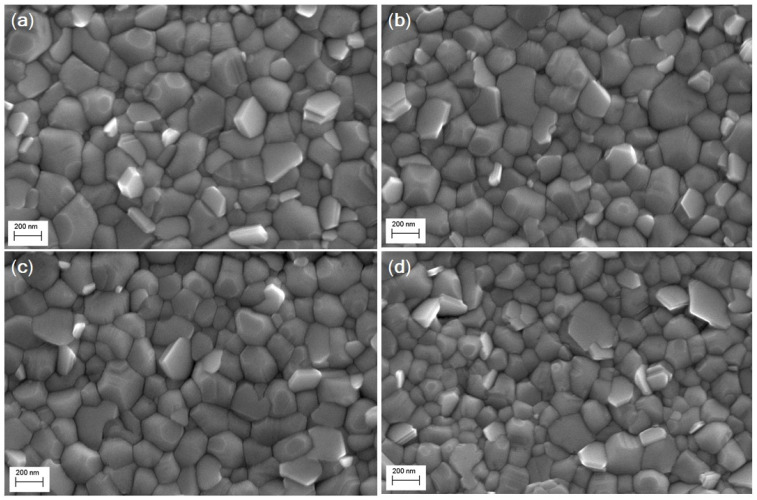
Top-view SEM images of the as-prepared CsMAFAPbIBr perovskite thin films, with and without MoS_2_ additive: (**a**) CsMAFAPbIBr; (**b**) CsMAFAPbIBr + MoS_2_ (5%); (**c**) CsMAFAPbIBr + MoS_2_ (10%); (**d**) CsMAFAPbIBr + MoS_2_ (20%).

**Figure 6 nanomaterials-11-01706-f006:**
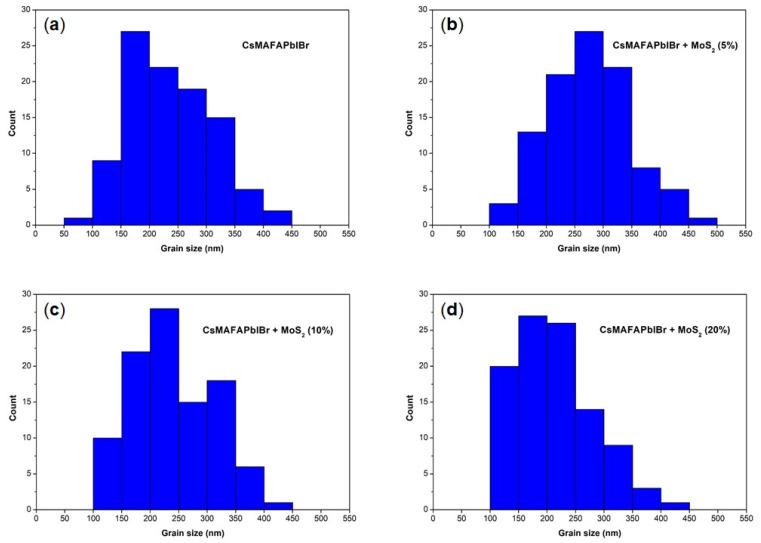
Particle size distribution histograms of the as-prepared CsMAFAPbIBr perovskite films, with and without MoS_2_ additive: (**a**) CsMAFAPbIBr; (**b**) CsMAFAPbIBr + MoS_2_ (5%); (**c**) CsMAFAPbIBr + MoS_2_ (10%); (**d**) CsMAFAPbIBr + MoS_2_ (20%).

**Figure 7 nanomaterials-11-01706-f007:**
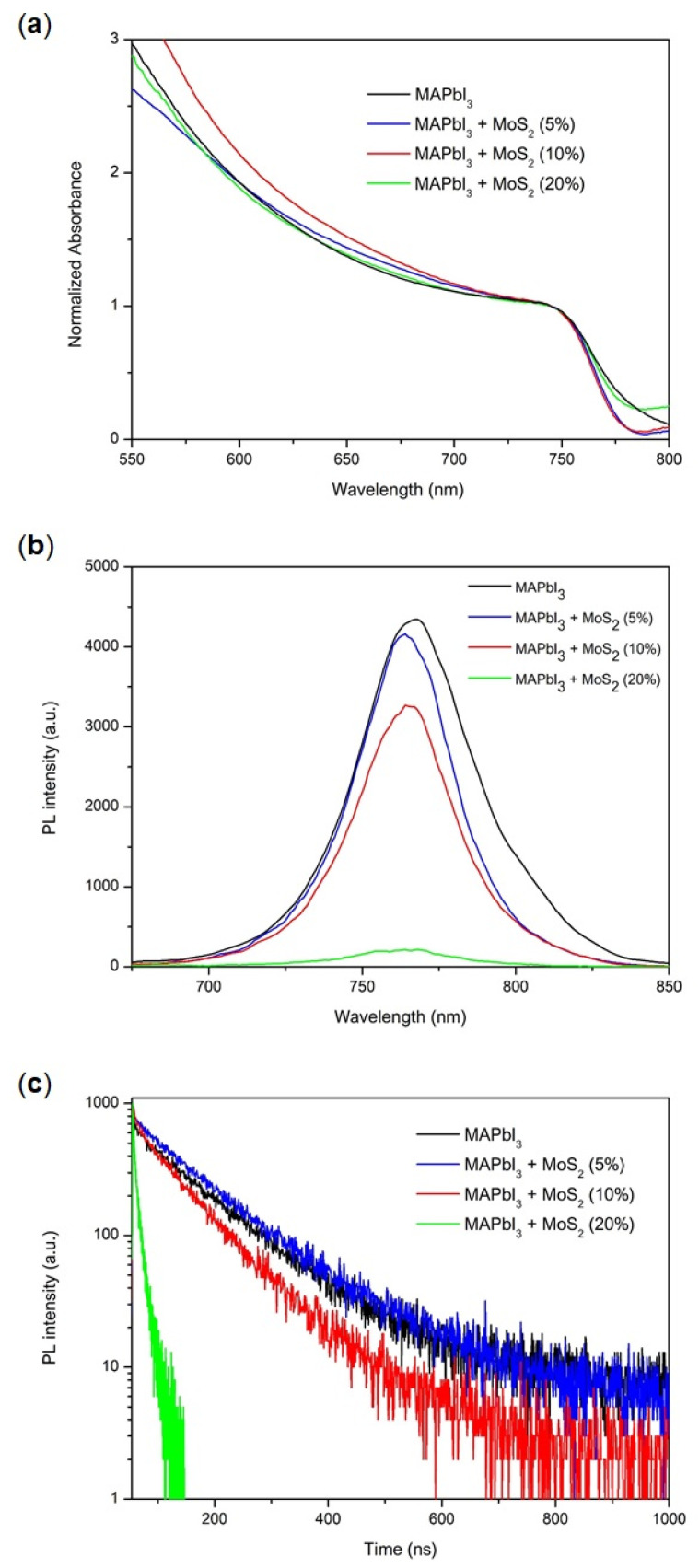
(**a**) UV-Vis, (**b**) steady-state photoluminescence (PL) and (**c**) time-resolved photoluminescence (TRPL) spectra of the MAPbI_3_ perovskite films, with and without MoS_2_ additive.

**Figure 8 nanomaterials-11-01706-f008:**
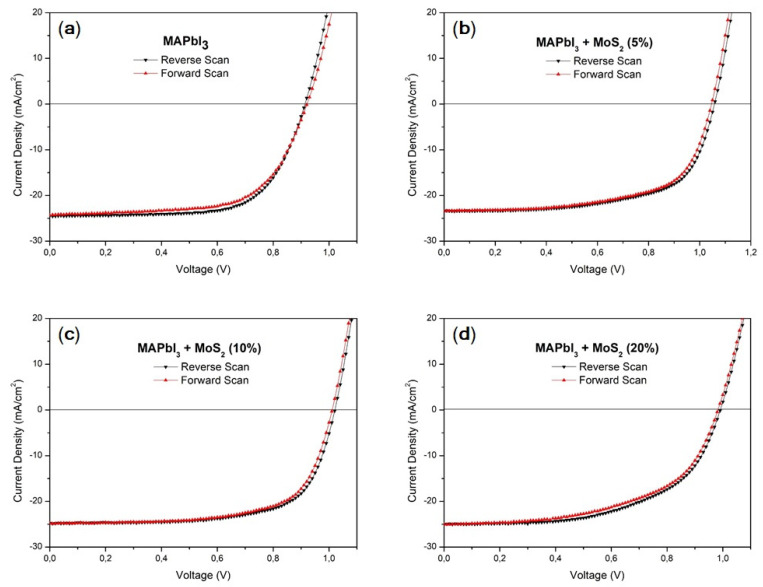
J-V curves of the MAPbI_3_ best-performing PSC: (**a**) MAPbI_3_; (**b**) MAPbI_3_ + MoS_2_ (5%); (**c**) MAPbI_3_ + MoS_2_ (10%); (**d**) MAPbI_3_ + MoS_2_ (20%).

**Figure 9 nanomaterials-11-01706-f009:**
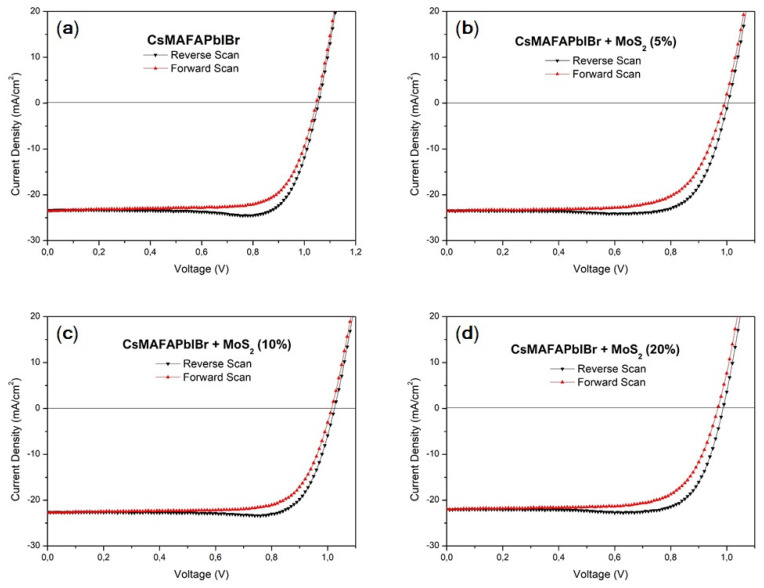
J-V curves of the CsMAFAPbIBr best-performing PSC: (**a**) CsMAFAPbIBr; (**b**) CsMAFAPbIBr + MoS_2_ (5%); (**c**) CsMAFAPbIBr + MoS_2_ (10%); (**d**) CsMAFAPbIBr + MoS_2_ (20%).

**Table 1 nanomaterials-11-01706-t001:** MAPbI_3_ film thicknesses, with and without MoS_2_ additive.

	Average Thickness (nm)
MAPbI_3_	422 ± 7
MAPbI_3_ + MoS_2_ (5%)	439 ± 15
MAPbI_3_ + MoS_2_ (10%)	436 ± 9
MAPbI_3_ + MoS_2_ (20%)	442 ± 20

**Table 2 nanomaterials-11-01706-t002:** Statistical analysis for the MAPbI_3_ grain size, with and without MoS_2_ additive.

	Mean (nm)	St. Dev. (nm)
MAPbI_3_	116.32	33.27
MAPbI_3_ + MoS_2_ (5%)	177.17	48.83
MAPbI_3_ + MoS_2_ (10%)	187.54	40.97
MAPbI_3_ + MoS_2_ (20%)	125.63	28.17

**Table 3 nanomaterials-11-01706-t003:** Statistical analysis for the CsMAFAPbIBr grain size, with and without MoS_2_ additive.

	Mean (nm)	St. Dev. (nm)
CsMAFAPbIBr	237.17	71.32
CsMAFAPbIBr + MoS_2_ (5%)	248.74	70.56
CsMAFAPbIBr + MoS_2_ (10%)	240.10	72.42
CsMAFAPbIBr + MoS_2_ (20%)	214.23	69.25

**Table 4 nanomaterials-11-01706-t004:** Photovoltaic characteristics of the MAPbI_3_ best-performing PSC and average values as well as standard deviations: reverse scan (black), forward scan (red).

	FF (%)	V_OC_ (V)	J_SC_ (mA cm^−2^)	PCE (%)
MAPbI_3_	67.4 (67.5 ± 1.8) 64.2 (66.9 ± 1.9)	0.91 (0.96 ± 0.04) 0.92 (0.96 ± 0.03)	24.49 (22.16 ± 1.65) 24.31 (22.20 ± 1.50)	15.09 (14.24 ± 0.60) 14.38 (14.20 ± 0.46)
MAPbI_3_ + MoS_2_ (5%)	64.5 (60.1 ± 8.6) 63.3 (58.7 ± 7.8)	1.06 (1.05 ± 0.01) 1.05 (1.03 ± 0.02)	23.45 (22.37 ± 0.78) 23.48 (22.25 ± 0.86)	16.01 (14.13 ± 2.31) 15.54 (13.51 ± 2.36)
MAPbI_3_ + MoS_2_ (10%)	68.9 (68.9 ± 1.1) 67.6 (67.5 ± 1.4)	1.02 (1.01 ± 0.01) 1.01 (1.00 ± 0.01)	24.76 (21.88 ± 2.05) 24.77 (21.91 ± 2.04)	17.42 (15.25 ± 1.55) 16.92 (14.85 ± 1.47)
MAPbI_3_ + MoS_2_ (20%)	67.4 (71.1 ± 3.2) 63.7 (67.8 ± 3.9)	0.92 (0.92 ± 0.01) 0.91 (0.92 ± 0.01)	21.46 (19.75 ± 2.00) 21.44 (19.71 ± 2.01)	13.30 (12.88 ± 1.59) 12.45 (12.30 ± 1.60)

**Table 5 nanomaterials-11-01706-t005:** Photovoltaic characteristics of the CsMAFAPbIBr best-performing PSC and average values as well as standard deviations: reverse scan (black), forward scan (red).

	FF (%)	V_OC_ (V)	J_SC_ (mA cm^−2^)	PCE (%)
CsMAFAPbIBr	78.6 (78.4 ± 4.9) 68.8 (70.3 ± 4.0)	1.03 (1.03 ± 0.01) 1.00 (1.03 ± 0.01)	23.37 (21.38 ± 1.34) 23.56 (20.58 ± 0.69)	18.89 (17.29 ± 1.13) 16.97 (14.89 ± 0.90)
CsMAFAPbIBr + MoS_2_ (5%)	78.2 (77.6 ± 4.8) 69.8 (70.6 ± 3.6)	1.01 (1.01 ± 0.03) 0.99 (1.01 ± 0.03)	23.47 (21.91 ± 1.66) 23.54 (21.86 ± 1.72)	18.45 (17.20 ± 1.19) 16.31 (15.45 ± 0.64)
CsMAFAPbIBr + MoS_2_ (10%)	81.3 (78.5 ± 4.3) 73.5 (71.8 ± 3.2)	1.03 (1.01 ± 0.02) 1.01 (1.00 ± 0.02)	22.62 (22.41 ± 1.41) 22.76 (22.46 ± 1.32)	18.85 (17.78 ± 0.81) 16.98 (16.19 ± 0.58)
CsMAFAPbIBr + MoS_2_ (20%)	78.8 (77.6 ± 3.2) 71.7 (70.7 ± 1.8)	1.01 (1.00 ± 0.02) 1.00 (0.99 ± 0.03)	22.08 (21.78 ± 0.76) 22.10 (21.81 ± 0.78)	17.59 (16.85 ± 0.63) 15.80 (15.21 ± 0.51)

## Data Availability

Data are fully available upon request.

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
