# Peer review of "Inclusion of 2D Transition Metal Dichalcogenides in Perovskite Inks and Their Influence on Solar Cell Performance"

_nanomaterials, 2021, doi:10.3390/nano11071706_

Round 1

Reviewer 1 Report

In this paper, the effect of two-dimensional additives on perovskite solar cells has been studied, but the content of the article is miscellaneous, the proof process is not clear enough, and the illustration of the picture is not strong enough.

1. The first phase of introduction is too verbose. It is suggested that the author select the main content of the article to focus on, instead of listing the relevant content one by one, which is not helpful to highlight the key points of the article.

2. The content of the second paragraph of the introduction is not clear. It is recommended that the author does not exhaust the history of development, but selects and briefly summarizes the papers that are instructive to the mechanism of the article.

3. In the last paragraph of the introduction, it is recommended to add a mechanism explanation instead of a simple description of the phenomenon.

4. In Figure 2c, why is the size of 10% additive the smallest? The author is suggested to explain.

5. When "It is believed that the excess of MoS2 additive can accelerate the formation of a disordered precursor phase during the early stage of the spin coating process that was previously shown to affect the perovskite film morphology/grain size even post – annealing" is mentioned in the paper, why does this phenomenon lead to the decrease of grain size? Is there no such phenomenon in a small number of additives?

6. When " with no spectral shift." is mentioned in the paper, however, a small amount of spectral shift can be seen in the figure, so it is suggested that the author give a reasonable explanation

7. In this paper, it is emphasized that the additives are flake morphology of electrochemical exfoliation, but whether they can maintain the morphology after dissolving in DMF and whether they are uniform flake morphology when affecting the growth of perovskite are not proved. Can the non-flake additives also achieve the same effect?

Author Response

Manuscript ID: 1239826

Title: Inclusion of 2D Transition Metal Dichalcogenides in Perovskite Inks

and their influence on solar cells performance.

We would like to thank all the reviewers for their appreciation to the work as well as to their careful evaluations/suggestions which allowed us to increase its soundness and potential impact.

The manuscript was accordingly amended, and revisions were reported in tracking mode.

Response to Reviewer 1

In this paper, the effect of two-dimensional additives on perovskite solar cells has been studied, but the content of the article is miscellaneous, the proof process is not clear enough, and the illustration of the picture is not strong enough.

  1. The first phase of introduction is too verbose. It is suggested that the author select the main content of the article to focus on, instead of listing the relevant content one by one, which is not helpful to highlight the key points of the article.
  2. The content of the second paragraph of the introduction is not clear. It is recommended that the author does not exhaust the history of development, but selects and briefly summarizes the papers that are instructive to the mechanism of the article.
  3. In the last paragraph of the introduction, it is recommended to add a mechanism explanation instead of a simple description of the phenomenon.

We thank the reviewer for his/her suggestions. We fully reviewed the introductive section of our work trying to improve its readability and soundness.

  1. In Figure 2c, why is the size of 10% additive the smallest? The author is suggested to explain.

Table in figure 2c (now 3c) report film thickness, it is true that among the additive containing films 10% is the thinner but the differences among samples are within the experimental error therefore we do not comment it in the text

  1. When "It is believed that the excess of MoS2 additive can accelerate the formation of a disordered precursor phase during the early stage of the spin coating process that was previously shown to affect the perovskite film morphology/grain size even post – annealing" is mentioned in the paper, why does this phenomenon lead to the decrease of grain size? Is there no such phenomenon in a small number of additives?

We thank the reviewer for his/her remark. We aggrege this discussion was confusing and distracting from the focus of nucleation. We removed it and replaced it with a discussion in line with the claims of the paper.

  1. When " with no spectral shift." is mentioned in the paper, however, a small amount of spectral shift can be seen in the figure, so it is suggested that the author give a reasonable explanation

The stoke shift is very small, we now briefly commented it

  1. In this paper, it is emphasized that the additives are flake morphology of electrochemical exfoliation, but whether they can maintain the morphology after dissolving in DMF and whether they are uniform flake morphology when affecting the growth of perovskite are not proved. Can the non-flake additives also achieve the same effect?

We thank the reviewer for this interesting question, we believe the shape of the TDM should be not affected by the solvation, therefore the flakes shapes should be retained in solution. The question related to alternative shapes impact on the nucleation is very interesting although not easy to address. In the case of Cs salts present in solution of mixed cation anion perovskite inks those evidently trigger the nucleation and most probably would be not presenting flakes shapes configuration being most probably spherical colloids or in alternative nanocrystals with a symmetry imposed by the elementary cells. We are planning further studied to address this issue which would find interesting following among the community.  

Reviewer 2 Report

Abstract : Good abstract, no comment. Introduction : Presentation : more paragraphs are advisable to bring out the structure of the intoduction at a glance. Some slight English error such as "and in perspective at low-cost" which is likely a literal translation form Italian (or French). "Low-cost" should not be hyphenanted. And "As most of the different processes occurring in the device, such as charge transport and diffusion lengths in the active media, are subjected to the morphology and crystallinity of the perovskite films." And "Additives inclusion in the perovskite" And so on : revise the grammar. A useful review of the interlayers and additives. Nevertheles it could do with more structure. Minor point : I would prefer dropping "ion" in "FA - formaminidium ion" and same for MA because FA is not specifically the ion, but the compound, and the same for MA. Experimental The sections are clear and succinct. The order should be materials, materials, preparation, fabrication, and conclude with characterisation. Re-arrange in consequence. Rename paragraph “characterisations” as “materials characterisations” (for example) to clearly differentiate from the concluding “Device fabrication and characterisation” The “device fabrication” should clearly show all the device structures fabricated. At present the information is present but unclear. I recommend adding device structures which can be referred to in the results section. Finally, I would recommend a separate “device characterisation” paragraph. Results and discussion Comment on structure : the results section starts with a repeat of elements already reported in the experimental section in the first paragraph, regarding preparation of materials. Suggestion : Drop “Preparation” from the title, and remove the first sentence, therefore starting a section “Characterisation” with “Scanning electron microscopy (SEM, fig. 1a)...” This is not a required change. This section discusses CH3NH3PbI3:MoS2 formation and characterisation. There is discussion of CsFAMAPbIBr fomation and characterisation in the “solar cells” section which should be moved here. CH3NH3PbI3:MoS2 film formation: Check the sentence “A preliminary check on the compatibility between MoS2 TMD sheets and the reaction environment in which the nucleation and growth process of the perovskite material is carried out confirmed the possibility of using these materials as additive in PSCs fabrication.” I think “these materials” should be “this material”, that is, MoS2. And “as additive” should therefore read “as an additive” Finally, “ in PSCs fabrication” should be “ in fabrication of PSCs” if you want the plural, or alternatively “ in PSC fabrication” in the the non-specific sense. Both forms are acceptable. I point this out in detail because there is some ambiguity for readers on what is meant by “these materials”. Correct “straightforward solubility”. What is meant is “good solubility” as in the preceding phrases. Check the phrase “coordination equilibria” which seems flawed. Perhaps “coordinated equilibria”. General comment : there is a tendency to propose hypotheses and justify them with references. While this is not wrong, it is nevertheless not a very strong approach. An example is the phrase starting “We hypothesize that the coordination equilibria...”. Since this is in part a question of writing style, no change is required. A suggestion on this point : the sentence above could be rewritten “This high solubility can be explained by coordinated equilibrium of lead cations and iodide ions [ref. 30]”. Change “grains size” to “grain size”. In general, carefully revise the grammar, the spelling, and use of plurals where they are not correct. No more details given on this point. The text says “This observation has relevant implications: it suggests a reduced defect density 236 at the grain boundaries, known to act as recombination sites between charge carriers” However the only evidence given is SEM images and greater grain size. While it can be agued that greater grain size reduces grain boundary recombination, it cannot be use to justify reduced defect density at the grain boundaries, which is how I understand the sentence. Perhaps what is intended is to claim reduced grain boundary recombination? Clarify this point and remove the claim “reduced defect density”, or justify it. Solar cells implementation Change the title since “Solar cells implementation” is ungrammatical. For example “Solar cell device characterisation”. Coming back to the earlier recommendation of “add figures showing device structures”, these should be referred to here. The section opens with a single long paragraph where the first sentences introduce the structures. The current structure is poor because the ETL and HTL materials are specified, but the ABX just labelled “perovskite” without clearly indiating that the materials in fig. 4b are mentioned. I suggest replacing “inverted p-i-n solar cell architecture composed of indium tin oxide (ITO)/polytriarylamine (PTAA)/perovskite/phenyl-C61-butyric acid methyl ester (PCBM)/bathocuproine (BCP)/aluminum (Al” with a phrase referring to the figures. This is of course a suggestion but the authors must clarify this section which currently is poorly presented as a single dense paragraph which does not allow the results to stand out, and minimises the potential impact of the article. Figure 4 must be separated into one figure containing the the light Ivs of fig. 4a which must be labelled independently, and a table containing the data in figure 4b. The table should avoid grey shading which is hard to view in some digital reproductions. Suggestion : the light IV data could be more usefully represented in the conventional photovoltaic light-IV format of negative current, and the zero current axis included to give the viewer a visual appeciation of the Voc. I would also myself eliminate the negative power region, even though I'm aware this is where hysterisis is most visible - but hysteresis beyond operating voltages is not very rich in informtion, whereas having a clearer view of the light current curve where the cell is delivering useful power is more helpful. Impact of MoS2 on triple catio ABX : Remove “tried” which is not good style. The work was performed, not merely attempted. Say “The approach reported in this work was extended to mixed halide...” Figure 5 must similarly be separated into a table showing the results and light Ivs for the triple cation ABX results added in order to allow equal comparison with the preceding perovkite data. As mentioned above, the materials synthesis and characterisation elements here (including the SEM figure) must be moved back to the same position as the preceding materials fabrication discussion, that is the earlier discussion of perovskite films referring to fig. 2 Conclusions No additional comment. Overall review comments : The current version shows shortcomings in presentation and in structure. The presentation consists of solid blocks of text which resemble a laboratory lab-book rather than a effort to convey results and analysis. In addition the structure of the paper is weak with scattered presentation of results, again reminiscent of a lab book rather than a well planned presentation of work done. Finally, the presentation of the data is below standard and asymmetric. There is missing data for some structures (light current-voltage curves), and a lack of clear presentation of the structures reported, and tables of data included in figures with graph, and with micrographs. All graph must be labelled using letters (e.g. 4a for the first light iv, 4b, 4c, 4d, and same for the micrographs currently in figure 5a). Though there is little analysis beyond references to the literature, this is interesting work. It can be published once the points made in this review has been attended to, which however consists of a major rewrite.

Author Response

Manuscript ID: 1239826

Title: Inclusion of 2D Transition Metal Dichalcogenides in Perovskite Inks

and their influence on solar cells performance.

We would like to thank all the reviewers for their appreciation to the work as well as to their careful evaluations/suggestions which allowed us to increase its soundness and potential impact.

The manuscript was accordingly amended, and revisions were reported in tracking mode.

Response to Reviewer 2

Abstract : Good abstract, no comment. Introduction : Presentation : more paragraphs are advisable to bring out the structure of the intoduction at a glance. Some slight English error such as "and in perspective at low-cost" which is likely a literal translation form Italian (or French). "Low-cost" should not be hyphenanted. And "As most of the different processes occurring in the device, such as charge transport and diffusion lengths in the active media, are subjected to the morphology and crystallinity of the perovskite films." And "Additives inclusion in the perovskite" And so on : revise the grammar. A useful review of the interlayers and additives. Nevertheles it could do with more structure. Minor point : I would prefer dropping "ion" in "FA - formaminidium ion" and same for MA because FA is not specifically the ion, but the compound, and the same for MA. Experimental The sections are clear and succinct. The order should be materials, materials, preparation, fabrication, and conclude with characterisation. Re-arrange in consequence. Rename paragraph “characterisations” as “materials characterisations” (for example) to clearly differentiate from the concluding “Device fabrication and characterisation” The “device fabrication” should clearly show all the device structures fabricated. At present the information is present but unclear. I recommend adding device structures which can be referred to in the results section. Finally, I would recommend a separate “device characterisation” paragraph. Results and discussion Comment on structure : the results section starts with a repeat of elements already reported in the experimental section in the first paragraph, regarding preparation of materials. Suggestion : Drop “Preparation” from the title, and remove the first sentence, therefore starting a section “Characterisation” with “Scanning electron microscopy (SEM, fig. 1a)...” This is not a required change. This section discusses CH3NH3PbI3:MoS2 formation and characterisation. There is discussion of CsFAMAPbIBr fomation and characterisation in the “solar cells” section which should be moved here. CH3NH3PbI3:MoS2 film formation: Check the sentence “A preliminary check on the compatibility between MoS2 TMD sheets and the reaction environment in which the nucleation and growth process of the perovskite material is carried out confirmed the possibility of using these materials as additive in PSCs fabrication.” I think “these materials” should be “this material”, that is, MoS2. And “as additive” should therefore read “as an additive” Finally, “ in PSCs fabrication” should be “ in fabrication of PSCs” if you want the plural, or alternatively “ in PSC fabrication” in the the non-specific sense. Both forms are acceptable. I point this out in detail because there is some ambiguity for readers on what is meant by “these materials”. Correct “straightforward solubility”. What is meant is “good solubility” as in the preceding phrases. Check the phrase “coordination equilibria” which seems flawed. Perhaps “coordinated equilibria”. General comment : there is a tendency to propose hypotheses and justify them with references. While this is not wrong, it is nevertheless not a very strong approach. An example is the phrase starting “We hypothesize that the coordination equilibria...”. Since this is in part a question of writing style, no change is required. A suggestion on this point : the sentence above could be rewritten “This high solubility can be explained by coordinated equilibrium of lead cations and iodide ions [ref. 30]”. Change “grains size” to “grain size”. In general, carefully revise the grammar, the spelling, and use of plurals where they are not correct. No more details given on this point. The text says “This observation has relevant implications: it suggests a reduced defect density 236 at the grain boundaries, known to act as recombination sites between charge carriers” However the only evidence given is SEM images and greater grain size. While it can be agued that greater grain size reduces grain boundary recombination, it cannot be use to justify reduced defect density at the grain boundaries, which is how I understand the sentence. Perhaps what is intended is to claim reduced grain boundary recombination? Clarify this point and remove the claim “reduced defect density”, or justify it. Solar cells implementation Change the title since “Solar cells implementation” is ungrammatical. For example “Solar cell device characterisation”. Coming back to the earlier recommendation of “add figures showing device structures”, these should be referred to here. The section opens with a single long paragraph where the first sentences introduce the structures. The current structure is poor because the ETL and HTL materials are specified, but the ABX just labelled “perovskite” without clearly indiating that the materials in fig. 4b are mentioned. I suggest replacing “inverted p-i-n solar cell architecture composed of indium tin oxide (ITO)/polytriarylamine (PTAA)/perovskite/phenyl-C61-butyric acid methyl ester (PCBM)/bathocuproine (BCP)/aluminum (Al” with a phrase referring to the figures. This is of course a suggestion but the authors must clarify this section which currently is poorly presented as a single dense paragraph which does not allow the results to stand out, and minimises the potential impact of the article. Figure 4 must be separated into one figure containing the the light Ivs of fig. 4a which must be labelled independently, and a table containing the data in figure 4b. The table should avoid grey shading which is hard to view in some digital reproductions. Suggestion : the light IV data could be more usefully represented in the conventional photovoltaic light-IV format of negative current, and the zero current axis included to give the viewer a visual appeciation of the Voc. I would also myself eliminate the negative power region, even though I'm aware this is where hysterisis is most visible - but hysteresis beyond operating voltages is not very rich in informtion, whereas having a clearer view of the light current curve where the cell is delivering useful power is more helpful. Impact of MoS2 on triple catio ABX : Remove “tried” which is not good style. The work was performed, not merely attempted. Say “The approach reported in this work was extended to mixed halide...” Figure 5 must similarly be separated into a table showing the results and light Ivs for the triple cation ABX results added in order to allow equal comparison with the preceding perovkite data. As mentioned above, the materials synthesis and characterisation elements here (including the SEM figure) must be moved back to the same position as the preceding materials fabrication discussion, that is the earlier discussion of perovskite films referring to fig. 2 Conclusions No additional comment. Overall review comments : The current version shows shortcomings in presentation and in structure. The presentation consists of solid blocks of text which resemble a laboratory lab-book rather than a effort to convey results and analysis. In addition the structure of the paper is weak with scattered presentation of results, again reminiscent of a lab book rather than a well planned presentation of work done. Finally, the presentation of the data is below standard and asymmetric. There is missing data for some structures (light current-voltage curves), and a lack of clear presentation of the structures reported, and tables of data included in figures with graph, and with micrographs. All graph must be labelled using letters (e.g. 4a for the first light iv, 4b, 4c, 4d, and same for the micrographs currently in figure 5a). Though there is little analysis beyond references to the literature, this is interesting work. It can be published once the points made in this review has been attended to, which however consists of a major rewrite.

We thank the reviewer for his/her careful reading of our work and for the precise suggestions. We went through all them and surely improved the manuscript in comparison to the previous submission. We basically went throughout a mayor rewriting of the work, we hope now it should be clearer and more useful to the community focussing on PSCs development.  

Round 2

Reviewer 1 Report

Based on revised MS, I think it is suitable for the Journal.

Author Response

Manuscript ID: 1239826

Title: Inclusion of 2D Transition Metal Dichalcogenides in Perovskite Inks and Their Influence on Solar Cells Performance.

Reviewer #1 (resubmission)

We would like to thank the reviewer for his/her appreciation to the work as well as for his/her careful evaluations/suggestions, which allowed us to increase its soundness and potential impact.

Reviewer 2 Report

Although there has been substantial improvement in the text, which is welcome, the review is nevertheless unsatisfactory and shows a lack of care.
The section “Response to reviewer” which I have access to only seems to contain the review, and this as a single block of text. This suggests a technical issue with the original review text having perhaps been pasted into the window with the wrong encoding - I cannot tell.

In detail :

Minor : “lowcost” : Previous revision indicated it should not be hyphenated. It should be two words, not a single word (despite this appearing frequently in advertising)

The second paragraph in the pdf file submitted by the authors is not comprehensible. This may be a format problem independent of the authors - I cannot tell.

This is repeated in subsequent sections of the revised document and render this review too time consuming and, in some sections, impossible to follow.

What is required in order to help the review proces is not simply  tracked change document as submitted, but a text file detailing all the changes the authors have made in response to the review.

The current revision is therefore insufficient.

In addition the figures are not satisfactory :

• Do not put “(2a)” on the constituent figures of figure 2. The “2” is redundant.
Replace with “(a)”.
Optional : Try and put a space between 2(a) and 2(b) (note in text it is correct to state the full “2(a)”.).

• Figure 3 : I believe my earlier revision requested that tabes are not to be combined with graphs.
If this was unclear : In this revision, remove the table “fig. 3c” and place as an independent figure.

• Figure 3 : There are four micrographs labelled “figure 3(d)” : separate this into four different figures.

• Figure three : following the earlier requests, I recommend separating this into two figures :
- Figure 3 containing the curent 3(a) and 3(b)
- Figure 4 containing 4(a) to 4(d), the micrographs which are currently all labelled 3(d).

• Figure 4 : this again is four micrographs but we only see a label 4a. Split into four as above.

• Remove further redundant letterings e.g. figure “(6a) => “(a)”, and so on.

• Figure 7 : remove “(7a)” : This is a single table, there is no need for lettering.

In the next revision, pay attention not just to the spirit of the review, but make an effort to write  clearly presented paper which shows attention to detail : the originl paper showed a lack of care. This review also shows a lack of care. If this lack of care contnues, the paper risks being rejected for wasting reviewer time.

Summary : 

- The format of the review is insufficient to review correctly. Author responses are not clear given the absence of  list of author responses. In addition the revised text is not in shape  allowing proper revision.

- The figures remain below standard.

As a consequence, the revision must be repeated, and in addition to comments in this revision, the first revision must also be re-examined, particularly in light of the figures which remain below standard. As a consequence, this merits a second major revision, the first major revision being impossible to judge adequately.

(Note : This text attached in pdf format in case of further formatting problems on the MDPI website)

Author Response

Manuscript ID: 1239826

Title: Inclusion of 2D Transition Metal Dichalcogenides in Perovskite Inks and Their Influence on Solar Cells Performance.

Since the reviewer ask to restructure also the first round reply, we report in this file a reply to the first and to the second submission report, we hope this would address the reviewer demand.

The uploaded revised text contains all the changes in a clear blue label

Reviewer #2 (first submission)

Abstract: Good abstract, no comment.

We thank the referee. In the final version, however, we restructured also the abstract to better mirror the reformulated claims of the work.

  1. Introduction, presentation: more paragraphs are advisable to bring out the structure of the introduction at a glance.

Answer: We modified the structure of the introduction accordingly.

  1. Some slight English error such as "and in perspective at low-cost" which is likely a literal translation form Italian (or French). "Low-cost" should not be hyphenanted. And "As most of the different processes occurring in the device, such as charge transport and diffusion lengths in the active media, are subjected to the morphology and crystallinity of the perovskite films." And "Additives inclusion in the perovskite" And so on: revise the grammar.

Answer: We modified the sentence “and in perspective at low-cost” with “and potentially at low cost”.

The sentence “As most of the different processes occurring in the device, such as charge transport and diffusion lengths in the active media, are subjected to the morphology and crystallinity of the perovskite films” has been deleted.

Finally, we modified the sentence:

Additives inclusion in the perovskite precursors solution are widely adopted aiming at efficient, stable, and hysteresis – free PSCs. Their method of function and influence on device performance are associated with the modulation of morphology of perovskite films14, stabilization of the active crystalline phase15, energy level alignment and suppression of non-radiative recombination in perovskite materials16

with:

“The inclusion of additives into the perovskite precursors solution is widely adopted as their presence impacts on the final morphology of perovskite films14, it stabilizes the active crystalline phase15, it tunes the energy level alignment between diverse material constituents and it suppresses non-radiative recombination in perovskite materials16”.

  1. A useful review of the interlayers and additives. Nevertheles it could do with more structure.

Answer: The structure of the text has been modified accordingly.

  1. Minor point: I would prefer dropping "ion" in "FA - formaminidium ion" and same for MA because FA is not specifically the ion, but the compound, and the same for MA.

Answer: The word “ion” has been deleted from the text.

  1. Experimental: The sections are clear and succinct. The order should be materials, materials preparation, fabrication, and conclude with characterization. Re-arrange in consequence. Rename paragraph “characterizations” as “materials characterizations” (for example) to clearly differentiate from the concluding “Device fabrication and characterization”. The “device fabrication” should clearly show all the device structures fabricated. At present the information is present but unclear. I recommend adding device structures which can be referred to in the results section. Finally, I would recommend a separate “device characterization” paragraph.

Answer: The experimental section has been now reorganized as required. The figures showing device structures have been now added (Fig. 1a,b), to which we refer in the experimental section and in the results section.

  1. Results and discussion, comment on structure: the results section starts with a repeat of elements already reported in the experimental section in the first paragraph, regarding preparation of materials. Suggestion: Drop “Preparation” from the title, and remove the first sentence, therefore starting a section “Characterisation” with “Scanning electron microscopy (SEM, fig. 1a)...” This is not a required change.

Answer: As suggested by the referee, the experimental results and characterizations are now organized in a single section named “Results and discussion”.

We modified the sentence:

MoS2 nanosheets were prepared by Liquid – Phase exfoliation.  Here, a typical methodology for the preparation of an exfoliated 2D MoS2 dispersion is performed (see experimental section). Scanning electron microscopy (SEM, Fig. 1a) image confirms the efficient exfoliation into few-layered MoS2 nanoflakes”

with:

“MoS2 nanosheets used in this work as additive in hybrid halide perovskite solutions were prepared by liquid – phase exfoliation (see experimental section). As first, the 2D material was characterized in order to confirm an efficient exfoliation process. Scanning electron microscopy (SEM, Fig. 2a) image shows few-layered MoS2 nanoflakes.”

  1. This section discusses CH3NH3PbI3:MoS2 formation and characterization. There is discussion of CsFAMAPbIBr formation and characterization in the “solar cells” section which should be moved here.

Answer: We thank the referee for the suggestion, we modified the text accordingly. SEM characterization of the CsMAFAPbIBr perovskite is now placed where suggested.    

  1. CH3NH3PbI3:MoS2 film formation: Check the sentence “A preliminary check on the compatibility between MoS2 TMD sheets and the reaction environment in which the nucleation and growth process of the perovskite material is carried out confirmed the possibility of using these materials as additive in PSCs fabrication.” I think “these materials” should be “this material”, that is, MoS2. And “as additive” should therefore read “as an additive” Finally, “ in PSCs fabrication” should be “ in fabrication of PSCs” if you want the plural, or alternatively “ in PSC fabrication” in the non-specific sense. Both forms are acceptable. I point this out in detail because there is some ambiguity for readers on what is meant by “these materials”.

Answer: We corrected the mentioned sentence as follow:

A preliminary check on the compatibility between MoS2 TMD sheets and the reaction environment in which the nucleation and growth process of the perovskite material take place confirmed the possibility of using this material as an additive for perovskite precursor solutions.”

  1. Correct “straightforward solubility”. What is meant is “good solubility” as in the preceding phrases. Check the phrase “coordination equilibria” which seems flawed. Perhaps “coordinated equilibria”. General comment: there is a tendency to propose hypotheses and justify them with references. While this is not wrong, it is nevertheless not a very strong approach. An example is the phrase starting “We hypothesize that the coordination equilibria...”. Since this is in part a question of writing style, no change is required. A suggestion on this point: the sentence above could be rewritten “This high solubility can be explained by coordinated equilibrium of lead cations and iodide ions [ref. 30]”.

Answer: We changed the text accordingly. The mentioned sentence is now replaced by:

In this environment, indeed, reasonably good solubility of the TMD is observed and no precipitation or phase separation occurs during the spin coating process. This high solubility can be explained by coordinated equilibrium of lead cations and iodide ions present in the perovskite precursors solution with MoS230.

  1. Change “grains size” to “grain size”. In general, carefully revise the grammar, the spelling, and use of plurals where they are not correct. No more details given on this point.

Answer: We did our best to correct the grammar mistakes as suggested by the referee.

  1. The text says “This observation has relevant implications: it suggests a reduced defect density at the grain boundaries, known to act as recombination sites between charge carriers”. However the only evidence given is SEM images and greater grain size. While it can be agued that greater grain size reduces grain boundary recombination, it cannot be use to justify reduced defect density at the grain boundaries, which is how I understand the sentence. Perhaps what is intended is to claim reduced grain boundary recombination? Clarify this point and remove the claim “reduced defect density”, or justify it.

Answer: We agree with this interesting observation. We replaced the mentioned sentence with:

This observation has relevant implications: it suggests a reduced extension of grain boundary region. This would prefigure a reduced charge loss, typically occurring at the grain region, turning in final device performances improvement”.

  1. Solar cells implementation: change the title since “Solar cells implementation” is ungrammatical. For example “Solar cell device characterization”. Coming back to the earlier recommendation of “add figures showing device structures”, these should be referred to here.

Answer: The experimental results and characterizations are now organized in a single section named “Results and discussion”. The figures showing device structures have been now added (Fig. 1a,b).

  1. The section opens with a single long paragraph where the first sentences introduce the structures. The current structure is poor because the ETL and HTL materials are specified, but the ABX just labelled “perovskite” without clearly indicating that the materials in fig. 4b are mentioned. I suggest replacing “inverted p-i-n solar cell architecture composed of indium tin oxide (ITO)/polytriarylamine (PTAA)/perovskite/phenyl-C61-butyric acid methyl ester (PCBM)/bathocuproine (BCP)/aluminum (Al” with a phrase referring to the figures. This is of course a suggestion but the authors must clarify this section which currently is poorly presented as a single dense paragraph which does not allow the results to stand out, and minimises the potential impact of the article.

Answer: We replaced the mentioned sentence with

Perovskite films with and without MoS2 additive were included in p-i-n solar cell architectures shown in Fig. 1.”

  1. Figure 4 must be separated into one figure containing the the light IVs of fig. 4a which must be labelled independently, and a table containing the data in figure 4b. The table should avoid grey shading which is hard to view in some digital reproductions. Suggestion: the light IV data could be more usefully represented in the conventional photovoltaic light-IV format of negative current, and the zero current axis included to give the viewer a visual appreciation of the Voc. I would also myself eliminate the negative power region, even though I'm aware this is where hysterisis is most visible - but hysteresis beyond operating voltages is not very rich in information, whereas having a clearer view of the light current curve where the cell is delivering useful power is more helpful.

Answer: We modified the light J-V curves and the tables accordingly.

  1. Impact of MoS2 on triple cation ABX: Remove “tried” which is not good style. The work was performed, not merely attempted. Say “The approach reported in this work was extended to mixed halide...” Figure 5 must similarly be separated into a table showing the results and light IVs for the triple cation ABX results added in order to allow equal comparison with the preceding perovskite data. As mentioned above, the materials synthesis and characterization elements here (including the SEM figure) must be moved back to the same position as the preceding materials fabrication discussion, that is the earlier discussion of perovskite films referring to fig. 2.

Answer: We modified the text accordingly in the SEM characterization section of the mixed cation-halide perovskite.

The light J-V curves and tables have been modified.

In addition, we have added the following paragraph:

PCE of the best device was above 18%, VOC was comprised between 1.01 and 1.03 V, FF above 78% and JSC above 23 mA/cm2for pristine perovskite and CsMAFAPbIBr:MoS2 (5%), while for MoS2 (10%) and MoS2 (20%) it slightly decreases to 22.62 and 22.08 mA/cm2, possibly due to the occurring of some nanosheets aggregation phenomena as observed for MAPbI3. However, these differences are negligible if we consider the very similar mean value and statistic distribution, that reflects the negligible differences in morphologies (Fig. 4) of the film observed with and without MoS2”.

  1. Conclusions: no additional comment. Overall review comments: The current version shows shortcomings in presentation and in structure. The presentation consists of solid blocks of text which resemble a laboratory lab-book rather than a effort to convey results and analysis. In addition the structure of the paper is weak with scattered presentation of results, again reminiscent of a lab book rather than a well planned presentation of work done. Finally, the presentation of the data is below standard and asymmetric. There is missing data for some structures (light current-voltage curves), and a lack of clear presentation of the structures reported, and tables of data included in figures with graph, and with micrographs. All graph must be labelled using letters (e.g. 4a for the first light iv, 4b, 4c, 4d, and same for the micrographs currently in figure 5a). Though there is little analysis beyond references to the literature, this is interesting work. It can be published once the points made in this review has been attended to, which however consists of a major rewrite.

Answer: We thank the reviewer for his/her careful reading of our work and for the precise suggestions. We went through all of them and we believe that the manuscript is now improved in comparison to the previous submission.

Reviewer #2 (resubmission)

  1. Although there has been substantial improvement in the text, which is welcome, the review is nevertheless unsatisfactory and shows a lack of care. The section “Response to reviewer” which I have access to only seems to contain the review, and this as a single block of text. This suggests a technical issue with the original review text having perhaps been pasted into the window with the wrong encoding - I cannot tell.

Answer: We are very sorry for that, we rewrote the response (above), listing all the modifications/corrections of the first submission. We hope that the response to review in the present form fulfils the reviewer request.

  1. In detail: Minor: “lowcost”: Previous revision indicated it should not be hyphenated. It should be two words, not a single word (despite this appearing frequently in advertising).

Answer: We modified the word “low-cost” with “low cost”.

  1. The second paragraph in the pdf file submitted by the authors is not comprehensible. This may be a format problem independent of the authors - I cannot tell. This is repeated in subsequent sections of the revised document and render this review too time consuming and, in some sections, impossible to follow.

Answer: We now present a clean version of the revised manuscript, with blue labelled changes.

  1. What is required in order to help the review process is not simply tracked change document as submitted, but a text file detailing all the changes the authors have made in response to the review. The current revision is therefore insufficient.

Answer: We now provide a response to the review in which all the changes are detailed. We also highlighted the changes in the revised version of the manuscript with the aim of making the revision easier.

  1. In addition the figures are not satisfactory. Do not put “(2a)” on the constituent figures of figure 2. The “2” is redundant. Replace with “(a)”. Optional: Try and put a space between 2(a) and 2(b) (note in text it is correct to state the full “2(a)”.). Figure 3: I believe my earlier revision requested that tabes are not to be combined with graphs. If this was unclear: in this revision, remove the table “fig. 3c” and place as an independent figure. Figure 3: There are four micrographs labelled “figure 3(d)”: separate this into four different figures. Figure 3: following the earlier requests, I recommend separating this into two figures: Figure 3 containing the current 3(a) and 3(b) Figure 4 containing 4(a) to 4(d), the micrographs which are currently all labelled 3(d). Figure 4: this again is four micrographs but we only see a label 4a. Split into four as above. Remove further redundant letterings e.g. figure “(6a) => “(a)”, and so on. Figure 7 : remove “(7a)”: This is a single table, there is no need for lettering.

Answer: We are very sorry for that, we corrected all the figures, light J-V curves and tables, accordingly. The J-V curves and SEM images are now labelled independently. Finally, the tables are separated from the graphs or micrographs. We hope the figures are now satisfactory.

  1. In the next revision, pay attention not just to the spirit of the review, but make an effort to write clearly presented paper which shows attention to detail: the originl paper showed a lack of care. This review also shows a lack of care. If this lack of care continues, the paper risks being rejected for wasting reviewer time. Summary: The format of the review is insufficient to review correctly. Author responses are not clear given the absence of list of author responses. In addition the revised text is not in shape allowing proper revision. The figures remain below standard. As a consequence, the revision must be repeated, and in addition to comments in this revision, the first revision must also be re-examined, particularly in light of the figures which remain below standard. As a consequence, this merits a second major revision, the first major revision being impossible to judge adequately.

Answer: We thank the reviewer for his/her suggestions. We went throughout the manuscript and substantially revising it. We also rewrite the response to the referees. We hope that the manuscript in the present form is clearer.

Round 3

Reviewer 2 Report

Abstract :

Clear, succinct, only a few minor issues with sentence structure (“based-upon”), no action needed, no further comment. Introduction : Well structured, the text perhaps on the long side. Check the spelling and grammar : points such as - “challenges for PSCs development” should be singular “challenges for PSC development”. - “TMDs sheets” similarly is singular : “TMD sheets”. And “foreseeing PSCs advancement”. And, it seems, most usages of “PSC”. Illustration : “The buses are red, but the bus colour scheme is in development”. Revise all uses of plurals in acronyms. - https://susy.mdpi.com/user/review/review/18184405/Gztrf3ux“impacts the final” - no “impacts on”. - “between diverse material constituents” : “diverse” is not the right word. I expect “the range of materials constituting the PSC” is what's means. Simply remove “diverse” it's not needed. These points are minor, but are raised because flawless papers have greater impact. Required change : define all acronyms. 2D-TMDs is not defined. Note definitions should be avoided in abstracts, since an abstract, being brief, does not generally need aronyms. Furthermore, any definitions in an abstract are not valid for the main text of a paper : they are separate documents. PCE is not defined. Comment : “LPE” defined as “liquid phase exfoliation” might be a little confusing since that's the standard acronym for liquid phase epitaxy. Try and find acronyms which don't clash. Furthermore, only use acronyms where essential, since acronyms do not, in general, make text more readable, rather the opposite. Experimental : Good detail. But as above, the grammar must be impoved. Sentences such as the ungrammatical “EDX spectrum of the MoS2 dispersion acquired on Jeol2100 (Jeol Ltd.) operated at 200 keV equipped with an Oxford Instruments 80mm2 silicon drift detector with a 10°holder tilt.” Results and discussion: Comment : The text is in parts verbose. See for example paragraph 3. Comment : It would be good to see quantitative analysis of the SEM results in fig. 3 by estimating grain sizes, perhaps using image analysis tools(open source solutions exist) for example for all pictures and including in table 1. One more : “are in the experimental errors” is wrong. “Are within experimental error” (singular) is correct. Structure suggestion : the results could be split into materials and device characterisation sub-sections. Comment : the results would benefit from QE measurements. Comment : “The J-V characteristics indicate that MoS2 sheets as an additive improve the device properties” is supported by the data, but only just. This is particularly interesting given that the mixed halide case shows no difference. There is a lack of analysis on this point. As a result, the reader is not convinced that the slight improvements seen which are close to the margin of error may not be due to secondary effects in the fabrication process. The paper would have greater impact if the reasons were explored, even if no full answer can be given without further investigation. Final note : The efficiencies reached are respectable but significantly below the state of the art. This might be mentioned. Conclusions : The conclusions could be written more precisely. One point that stands out is “larger grains and reduced grain boundary recombination” : this is purely phenomenological, since no results supporting this are provided other than SEM images and visual inspection. The paper would be improved by mitigating claims or by adding defect characterisation justifying the point. The discussion of lack of improvement for mixed halide does not add significantly to the conclusions. It states simply that different materials nucleate differently and therefore may or may not benefit from TMD additives. There is a lack of analysis on this point. Overall comments : The paper is in good shape regading content. It is well referenced, contains a good level of detail on the fabriation and characterisation. Greater analysis would give this paper greater impact. At present it reads like a paper reporting results of trial and error which work in one case and not in another, but is lacking thought and analysis to understand why and most importantly does not therefore convince the reader that this additive can have a significant impact. The originality and merit is therefore less than it might be. The language lets the paper down. Though none of the issues picked up are critical, they are numerous enough to require a revision of the entire text. In addition, the text is in some sections verbose rather than written to get the message across efficiently. The fabrication section in particular is written in telegraphic and ungrammatical style in incomplete sentences. These are minor revisions.

Author Response

Manuscript ID: 1239826

Title: Inclusion of 2D Transition Metal Dichalcogenides in Perovskite Inks and Their Influence on Solar Cells Performance.

We thank the reviewer for his/her appreciation to the work as well as to his/her careful evaluations/suggestions. The manuscript was accordingly amended, and revisions are highlighted in the text and listed below.

Reviewer #2 (Round 3)

  1. Abstract: clear, succinct, only a few minor issues with sentence structure (“based-upon”), no action needed, no further comment.

Answer: We corrected the mentioned sentence as follow:

“Organic – inorganic hybrid perovskite materials have raised great interest in recent years due to their excellent optoelectronic properties, which promise stunning improvements in photovoltaic technologies.”

  1. Introduction: Well structured, the text perhaps on the long side. Check the spelling and grammar: points such as - “challenges for PSCs development” should be singular “challenges for PSC development”. - “TMDs sheets” similarly is singular: “TMD sheets”. And “foreseeing PSCs advancement”. And, it seems, most usages of “PSC”. Revise all uses of plurals in acronyms. “Impacts the final” - no “impacts on”. - “Between diverse material constituents”: “diverse” is not the right word. I expect “the range of materials constituting the PSC” is what's means. Simply remove “diverse”, it's not needed. These points are minor, but are raised because flawless papers have greater impact.

Answer: We have revised all the acronyms in the main text accordingly.

Finally, we modified the sentence:

“The inclusion of additives into the perovskite precursors solution is widely adopted as their presence impacts on the final morphology of perovskite films14, it stabilizes the active crystalline phase15, it tunes the energy level alignment between diverse material constituents and it suppresses non-radiative recombination in perovskite materials16.”

with:

“The inclusion of additives into the perovskite precursor solution is widely adopted as their presence impacts the final morphology of perovskite films14, it stabilizes the active crystalline phase15, it tunes the energy level alignment between material constituents and it suppresses non-radiative recombination in perovskite materials16.”

  1. Required change: define all acronyms. 2D-TMDs is not defined. Note definitions should be avoided in abstracts, since an abstract, being brief, does not generally need acronyms. Furthermore, any definitions in an abstract are not valid for the main text of a paper: they are separate documents. PCE is not defined. Comment: “LPE” defined as “liquid phase exfoliation” might be a little confusing since that's the standard acronym for liquid phase epitaxy. Try and find acronyms which don't clash. Furthermore, only use acronyms where essential, since acronyms do not, in general, make text more readable, rather the opposite.

Answer: We thank the referee for the suggestion, all acronyms in the main text have been defined. Acronyms and definitions in the abstract have been removed. As suggested by the referee, the acronym “LPE” has been deleted.

  1. Experimental: Good detail. But as above, the grammar must be improved. Sentences such as the ungrammatical “EDX spectrum of the MoS2 dispersion acquired on Jeol2100 (Jeol Ltd.) operated at 200 keV equipped with an Oxford Instruments 80mm2 silicon drift detector with a 10° holder tilt.”

Answer: We corrected the grammar of the experimental section. The sentences:

“The SEM imaging was performed by the MERLIN Zeiss SEM FEG instrument at an accelerating voltage of 5 kV, using an in – lens detector. HRTEM images acquired on an FEI Titan (Thermo Fisher Scientific Inc.) operated at an acceleration voltage of 300 keV. EDX spectrum of the MoS2 dispersion acquired on Jeol2100 (Jeol Ltd.) operated at 200 keV equipped with an Oxford Instruments 80mm2 silicon drift detector with a 10° holder tilt.”

have been replaced by:

“The SEM imaging was performed by a MERLIN Zeiss SEM FEG instrument at an accelerating voltage of 5 kV, using an in – lens detector. The particle size distribution was estimated using the open source ImageJ software, by measuring the major axis of 100 perovskite grains for each sample. HRTEM images were acquired on an FEI Titan (Thermo Fisher Scientific Inc.) operating at an acceleration voltage of 300 keV. EDX spectrum of the MoS2 dispersion was acquired on Jeol2100 (Jeol Ltd.) operating at 200 keV and equipped with an Oxford Instruments 80mm2 silicon drift detector with a 10° holder tilt.”

  1. Results and discussion: Comment: The text is in parts verbose. See for example paragraph 3. Comment: It would be good to see quantitative analysis of the SEM results in fig. 3 by estimating grain sizes, perhaps using image analysis tools (open source solutions exist) for example for all pictures and including in table 1.

Answer: We agree with the referee and we thank him/her for the suggestion. We have now added the particle size distribution and the statistical analysis for the grain size of both studied perovskites (Fig. 4 and Table 2 for the MAPbI3 perovskite, Fig. 6 and Table 3 for the CsMAFAPbIBr perovskite).

The paragraphs:

“SEM images (Fig. 3b,c) show the MoS2 additive influence on the MAPbI3 perovskite grain size: the addition of additive into perovskite precursors solutions, at low – intermediate concentrations (5% and 10%), results in the formation of larger – on average – grain. This observation has relevant implications: it suggests a reduced extension of grain boundary region. This would prefigure a reduced charge loss, typically occurring at the grain region, turning in final device performances improvement.

There are several factors that can affect the perovskite morphology including the intrinsic properties of the material itself, the deposition methods, the existence of impurities, the surface energy of the substrate and the application of post – treatment31. MoS2 additive seems to be able to regulate the film morphology by acting on the crystal growth and by altering the colloid distribution in the perovskite precursors31, which results in high quality pinhole – free perovskite films with larger grain size and filled grain boundaries33,34.  Since the crystal growth rate is relatively fast as a function of the solution supersaturation, reaching a high nucleation rate before the onset of crystal growth is mandatory to improve the perovskite film coverage. The heterogeneous nucleation mechanism in addition provides fewer nucleation sites, if compared to the homogeneous nucleation, leading to the formation of larger crystalline domains34,35. This interesting observation is extended here to the deposition of perovskite upon organic PTAA substrates, showing that the TMDs could also affect this kind of device, which would be ideally useful for flexible, light, and portable PSCs. Noticeably, at 20% MoS2 additive concentration, the effect on the grain size is lost as these films exhibit similar grain sizes to pristine perovskite. The effect of such MoS2 high concentration is ascribed to an increased number of heterogeneous nucleation sites that would impair the formation of large grains during the perovskite crystal growth. This is suggested by the SEM image in Fig. 3d. A morphology characterized by smaller grains, as the one recorded for the 20% MoS2additive:perovskite film, is in general associated to poor charge transport and collection in PSCs, which are limited by the inter – grain boundary recombination losses36.

The approach reported in this work was extended to mixed cation-halide CsMAFAPbIBr perovskite that is among the best performing perovskite material for solar cell. Surprisingly, we found that the inclusion of MoS2 additive into triple-cation did not significantly affect the morphology of perovskite films (Fig. 4), all the perovskite samples indeed exhibit a uniform and compact surface with similar grain sizes. This evidence can be mainly ascribed to the marginal role of the additive in influencing and controlling the crystallization dynamics of the CsMAFAPbIBr perovskite material. We suppose that in this case, due to the lower solubility of cesium salts, those providing heterogeneous nucleation seeds would promote the heterogenic crystal growth35

have been changed with:

“SEM images (Fig. 3b,c) show the MoS2 additive’s influence on the MAPbI3 perovskite grain size: the addition of the additive into the perovskite precursor solutions, at low – intermediate concentrations (5% and 10%), results in the formation of larger grains. The particle size analysis of these images (Fig. 4, Table 2) reveals a significant increase indeed in the average grain size, from 116 nm for the pristine sample to 177 nm and 187 nm for the MoS2 (5%) and MoS2 (10%) perovskite samples, respectively. Moreover, the 10% MoS2 additive sample has the narrowest grain size distribution, with more than 50% of particles located in a small range from 160 nm to 220 nm. The implication of this observation is a reduced extension of the grain boundary region. This prefigures a reduced charge loss, typically occurring at the grain boundary region, giving rise to an improvement in final device performance.

There are several factors that can affect the perovskite’s morphology including the intrinsic properties of the material itself, the deposition method, the presence of impurities, the surface energy of the substrate, and the application of post – treatments31. The MoS2 additive seems to be able to regulate film morphology by acting upon the crystal growth and by altering the colloid distribution in the perovskite precursors31. This results in high quality, pinhole – free perovskite films with larger grain size and filled grain boundaries33,34.  Since the crystal growth rate is relatively fast and is a function of solution supersaturation, reaching a high nucleation rate before the onset of crystal growth is required to improve perovskite film coverage. The heterogeneous nucleation mechanism in addition provides fewer nucleation sites, when compared to homogeneous nucleation, leading to the formation of larger crystalline domains34,35. This interesting observation is extended here to the deposition of perovskite onto organic PTAA substrates, showing that TMDs can also affect this kind of device, which would be ideally suited for flexible, light, and portable PSCs. Noticeably, at 20% MoS2 additive concentration, the effect on the grain size is lost as these films exhibit similar grain sizes and particle size distribution to pristine perovskite (Fig. 4, Table 2). The effect of such MoS2 high concentration is ascribed to an increased number of heterogeneous nucleation sites that would impair the formation of large grains during the perovskite crystal growth. This is suggested by the SEM image in Fig. 3d. A morphology characterized by smaller grains, as the one recorded for the 20% MoS2 additive perovskite film, is in general associated with poor charge transport and collection in PSCs, which are limited by inter – grain boundary recombination losses36.

The approach reported in this work was extended to mixed cation-halide CsMAFAPbIBr perovskite that is among the best performing perovskite material for solar cells. Surprisingly, we found that the inclusion of MoS2 additive into triple-cation did not significantly affect the final morphology of perovskite films (Fig. 5), as all the samples exhibit a uniform and compact surface with similar grain size distribution (Fig. 6, Table 3). This evidence can be mainly ascribed to the marginal role of the additive in influencing and controlling the crystallization dynamics of the CsMAFAPbIBr perovskite material. We suppose that in this case, the lower solubility cesium salts act as heterogeneous nucleation seeds, which promote heterogenic crystal growth35.”

  1. One more: “are in the experimental errors” is wrong. “Are within experimental error” (singular) is correct.

Answer: We changed the text accordingly. The mentioned sentence is now replaced by:

“Small differences among the three samples are within experimental error.”

  1. Structure suggestion: the results could be split into materials and device characterisation sub-sections.

Answer: We thank the reviewer for the suggestion. The “Results and discussion” section is now split into two sub-sections, named “Properties and characterization of materials” and “Photovoltaic performances”.

  1. Comment: the results would benefit from QE measurements.

Answer: This is an interesting suggestion, however in order to perform such measurements we would need to repeat the fabrication and characterization of all the presented devices, this would require a long time also due to the current pandemic situation.

  1. Comment: “The J-V characteristics indicate that MoS2 sheets as an additive improve the device properties” is supported by the data, but only just. This is particularly interesting given that the mixed halide case shows no difference. There is a lack of analysis on this point. As a result, the reader is not convinced that the slight improvements seen which are close to the margin of error may not be due to secondary effects in the fabrication process. The paper would have greater impact if the reasons were explored, even if no full answer can be given without further investigation.

Answer: We believe that in the last version of the manuscript these differences have been more solidly underlined, the analysis of the SEM images shows interesting differences among the mixed halide cation and the pristine perovskite upon additive addition with respect to the average grain dimensions, and these observations soundly add to our claims. The differences among the PV performances in the case of MAPbI3 device do not fall within an error distribution, being above 15%, but those are clearly attributable to the film morphologies differences.

  1. Final note: The efficiencies reached are respectable but significantly below the state of the art. This might be mentioned.

Answer: We thank the referee for this observation. It is certainly true that the recorded efficiencies are below the state of the art, but we believe that a direct comparison between different architectures, which employ different charge transporting layers, is incorrect. Moreover, the use of additives in well-defined concentrations, strictly related to the chemical composition of the active material and the mixed solvents used, further prevents comparison with other materials.

  1. Conclusions: The conclusions could be written more precisely. One point that stands out is “larger grains and reduced grain boundary recombination”: this is purely phenomenological, since no results supporting this are provided other than SEM images and visual inspection. The paper would be improved by mitigating claims or by adding defect characterization justifying the point.

Answer: The particle size distribution and the statistical analysis for the grain size now support the mentioned claim. While it can certainly be said that greater grain size reduces the extension of grain boundary region, we modified the sentence as follow:

“Our findings suggest that MoS2 incorporation (10% V/V) into a MAPbI3 perovskite photoactive layer results in high – quality films with larger grains and optimized morphology, suitable for device integration.”

  1. The discussion of lack of improvement for mixed halide does not add significantly to the conclusions. It states simply that different materials nucleate differently and therefore may or may not benefit from TMD additives. There is a lack of analysis on this point.

Answer: We modified the sentence following the referee suggestions:

“Conversely, for the mixed cation-halide perovskite no improvements were observed with the addition of the additive to the precursors ink, in the case of Cs containing solution the nucleation would in fact be triggered by insoluble Cs salts seeds. Our results confirm how the nucleation process differs for different perovskite precursors composition and so the process would be differently influenced by the presence of additives.”

was change with:

“Conversely, for the mixed cation-halide perovskite no improvements were observed with the addition of the additive to the precursor ink. Our results confirm how the nucleation process differs for distinct perovskite precursor compositions and so, the process is influenced differently by the presence of additives.”

  1. Overall comments: The paper is in good shape regarding content. It is well referenced, contains a good level of detail on the fabrication and characterization. Greater analysis would give this paper greater impact. At present it reads like a paper reporting results of trial and error which work in one case and not in another but is lacking thought and analysis to understand why and most importantly does not therefore convince the reader that this additive can have a significant impact. The originality and merit is therefore less than it might be. The language lets the paper down. Though none of the issues picked up are critical, they are numerous enough to require a revision of the entire text. In addition, the text is in some sections verbose rather than written to get the message across efficiently. The fabrication section in particular is written in telegraphic and ungrammatical style in incomplete sentences. These are minor revisions. 

Answer: We thank the reviewer for his/her appreciation to the work as well as his/her helpful suggestions. We have substantially revised and organized the entire manuscript as required, we have added further characterization of the materials, we corrected all the tables, figures and J-V curves accordingly, trying to satisfy every request. Finally, we had the manuscript checked by a native English-speaking colleague. We hope that the revised version of manuscript is now satisfactory.